# Bruton’s Tyrosine Kinase Inhibitors: The Next Frontier of B-Cell-Targeted Therapies for Cancer, Autoimmune Disorders, and Multiple Sclerosis

**DOI:** 10.3390/jcm11206139

**Published:** 2022-10-18

**Authors:** Neeta Garg, Elizabeth Jordan Padron, Kottil W. Rammohan, Courtney Frances Goodman

**Affiliations:** Miller School of Medicine, University of Miami, Miami, FL 33136, USA

**Keywords:** BTKi, Bruton tyrosine kinase, multiple sclerosis, autoimmune disease, cancer, B-cell, treatment

## Abstract

Bruton’s tyrosine kinase (BTK) is an important protein belonging to the tyrosine kinase family that plays a key role in the intracellular signaling and proliferation, migration, and survival of normal and malignant B-lymphocytes and myeloid cells. Understanding the role of BTK in the B-cell signaling pathway has led to the development of BTK inhibitors (BTKi) as effective therapies for malignancies of myeloid origin and exploration as a promising therapeutic option for other cancers. Given its central function in B-cell receptor signaling, inhibition of BTK is an attractive approach for the treatment of a wide variety of autoimmune diseases that involve aberrant B-cell function including systemic lupus erythematosus (SLE), rheumatoid arthritis (RA), and multiple sclerosis (MS). Here, we review the role of BTK in different cell signaling pathways, the development of BTKi in B-cell malignancies, and their emerging role in the treatment of MS and other autoimmune disorders.

## 1. Introduction

Tyrosine kinases (TK) play a role in signal transmission from various cell surface receptors, including diverse leukocyte antigen and cytokine receptors, thereby mediating the recruitment and activation of different leukocyte populations [1]. Bruton’s tyrosine kinase (BTK) is a member of the Tec family of cytoplasmic, non-receptor tyrosine kinases that are mainly expressed on hematopoietic cells including B cells, myeloid cells, and platelets but not in T, plasma, or natural killer cells. BTK is critical to intracellular signaling in B cells and cells of the myeloid lineage including monocytes and macrophages [2]. BTK is a key enzyme in the B-cell receptor (BCR) signal transduction pathway and an essential signaling molecule downstream of BCR that is central to regulation of B-cell development, proliferation, activation, and differentiation into memory B cells and antibody producing plasma cells (Figure 1). Of note, after differentiating into plasma cells BTK is no longer active in these cells. BTK contributes to oncogenic signaling that is essential for proliferation and survival of malignant cells in various B-cell malignancies [3]. BTK is also involved in other cell signaling pathways including NFκβ (nuclear factor κappa β), NFAT (nuclear factor of activated T cells), and MAPK (mitogen-activated protein kinase) pathways (Figure 2) [4,5]. BTK inhibition results in inactivation of downstream cellular pathways including ERK (extracellular signal-regulated kinase), PKB (protein kinase B, also known as Akt), and NFκβ causing breakdown of cell-to-cell communication through the disruption of signaling via chemotaxis and adhesion. Eventually, this leads to disruption of DNA synthesis and apoptosis within B cells. Given BTK’s role in B-cell and myeloid cell homeostasis, BTK inhibition has been established as an effective strategy for treatment of B-cell leukemia and other hematological malignancies.

As BTK is involved in B-cell and other cell signaling pathways, BTKi also appear to be promising drugs for treatment of autoimmune diseases with aberrant B-cell function such as SLE, RA, and MS [6,7]. BTK modulates cell signaling pathways of B cells and other myeloid cells including monocytes, macrophages and microglia that are important in the pathogenesis of MS. BTK inhibition can potentially downregulate inflammation and promote remyelination through suppression of microglial activation in the central nervous system (CNS) [8,9,10]. The importance of B cells as mediators of inflammation in MS has been well established and supported by recent successful trials of B-cell depleting agents [11,12,13]. The mechanism of action of B-cell depleting therapies is believed to involve the non-specific depletion of peripheral memory B cells leading to a decrease in antigen presentation to T cells and cytokine production. Although B-cell depletion is a highly effective therapeutic strategy, the risk of long-term immune suppression resulting from chronic usage of B-cell depleting therapies remains a concern [14]. This has prompted search for more selective B-cell targeted agents which potentially have a better safety profile.

## 2. Historic Background/Mechanism of Action/Molecular Effects

BTK is named after Ogden Bruton, who first discovered x-linked agammaglobulinemia (XLA) in 1952 [15]. BTK was first identified in 1993 as a signaling molecule affected in XLA, an inherited immunodeficiency disease characterized by absence of peripheral mature B cells and serum immunoglobulins due to defect in B-cell development [16,17]. A few rare cases of autoimmune disease in XLA patients have been described. Up to 90% of patients diagnosed with early-onset XLA carry mutations in the *BTK* gene located on the X chromosome [17]. Mutations in the BTK gene lead to less severe X-linked immunodeficiency in mice [18].

## 3. Role of BTK in B-Cell Development and Function

BTK belongs to a five-member family of Tec non-receptor tyrosine kinases composed of tyrosine kinase expressed in hepatocellular carcinoma (TeC), BTK, interleukin-2-inducible T-cell kinase (ITK), bone marrow-expressed kinase (BMX), and tyrosine protein kinase/redundant-resting lymphocyte kinase (TXK/RLK); these kinases are mainly expressed on hematopoietic cells [19]. BTK plays a central role in intracellular signaling in B cells via a signal transduction pathway downstream of BCR [2]. BCR is a transmembrane signaling complex expressed on the B-cell surface, composed of immunoglobulin heavy (IGH) and light (IGL) chains (antigen recognition unit), co-receptor proteins Igα CD79A, Igβ CD79B, and CD19 (signaling unit). BCR is expressed on most B cells including normal and malignant B-lymphocytes and is involved in B-cell growth, differentiation, and function. BTK regulates several processes in B-cell development and activation, in particular the pre-BCR checkpoint through regulation of IL-7 responsiveness, activation of λ light chain gene rearrangements, and differentiation into memory or plasma cells (Figure 1) [20,21]. The BTK-dependent signaling cascade following BCR stimulation involves a series of enzymatic activation (Figure 2) that eventually results in altered expression of genes that regulate B-cell differentiation, proliferation, DNA repair, survival, cytoskeleton remodeling, and migration [22,23]. After initial stimulation of BCR through antigen recognition, BTK is activated by tyrosine protein kinases, LYN (Lck/Yes novel kinase) and SYK (spleen tyrosine kinase), this is followed by activation of 1-phosphatidylinositol-4,5-biphosphate phosphodiesterase gamma-2 (PLC-γ2) and phosphorylation and full activation of BTK. The activation of PLC-γ2 and BTK leads to a variety of downstream effects that regulate activation of nuclear factor κ-light-chain-enhancer of activated B cells (NF-κB), nuclear factor of activated T-cells (NFAT), and extracellular signal-regulated kinases (ERK) pathways (Figure 2) [24,25,26,27]. BTK also activates AKT (also known as protein kinase B), another protein kinase important for survival signaling through activation of phosphoinositide 3-kinase/phosphatidylinositol (3,4,5)-trisphosphate (PI3K/PIP3) [28]. BTK therefore is essential for B-cell survival with a high rate of B-cell apoptosis when it is deficient or absent correlating with strongly reduced BCR-mediated induction of the anti-apoptotic protein, B-cell lymphoma-extra-large [29]. In the absence of BTK, B cells enter the early G1 phase but fail to enter into S phase of the cell cycle. The cell cycle is rendered incapacitated because of the lack of cyclin D2 expression. Furthermore, BTK is highly involved in B-cell differentiation into a B-1 cell phenotype, which is a part of the innate immune system [30]. Importantly, BTK can modulate signaling, acting as a “rheostat” rather than an “on-off” switch; thus, overexpression leads to autoimmunity while decreased levels improve autoimmune disease outcomes. Autoreactive B cells depend upon BTK for survival to a greater degree than normal B cells with increased BTK expression correlating with autoantibody production in patients with systemic autoimmune diseases [31]. BTK is also involved in other B-cell-intrinsic signaling pathways essential for cellular survival and propagates BCR responses to stimulation via CD40, chemokine receptor, adhesion molecules, Toll-like receptor (TLR) and Fc receptors [32,33].

**Figure 1 jcm-11-06139-f001:**
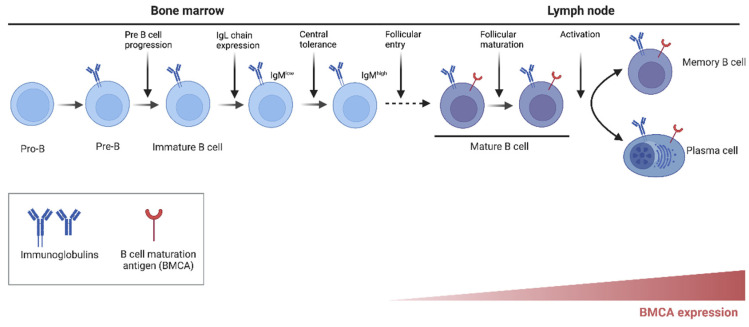
Role of BTK in B-cell development and activation from Pro-B cells (first expression of BTK) to fully differentiated memory B cells or plasma cells. The key steps including progression to pre-B cells, expression of immunoglobulin chains, follicular maturation, and differentiation into memory and plasma cells are indicated by vertical arrows [21].

**Figure 2 jcm-11-06139-f002:**
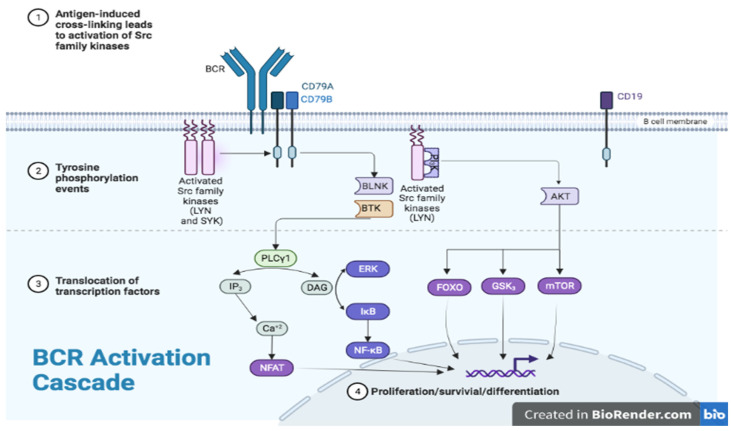
Role of BTK in antigen dependent BCR signaling pathway that regulates B-cell proliferation and survival [23].

## 4. BTK in Other Myeloid Cells

BTK is also expressed in cells of erythroid and myeloid lineages including erythroblast, reticulocytes, and erythrocytes, as well as monocytes, mast cells, macrophages, granulocytes, dendritic cells, and osteoclasts. BTK is important for the function and activation of these cells by controlling production of cytokines and inflammatory mediators and cytokine-mediated intracellular signaling [5]. Platelet and myeloid cell development have shown to be dependent on TLR-mediated activation; mast cells utilize the high-affinity IgE receptor, known as FcεR, and macrophages use the high-affinity IgG receptor and FcyRI, both Toll-like receptors [34,35,36]. Furthermore, BTK is also involved in the osteoclast differentiation signaling pathway via activation of NF-kB (RANK) and platelet signaling via the collagen receptor glycoprotein VI [37,38].

## 5. BTK and T Lymphocytes

T lymphocytes are devoid of BTK activity but are influenced indirectly through the effects of BTK inhibitors on the myeloid system. BTK and its family kinases are involved in immune tolerance, creation and regulation of tumor microenvironment, and tumor immune-escape mechanisms. These mechanisms include CD4 Th1 to Th2 lymphocyte shift with resultant reduction in CD8+ cytotoxic T cells, upregulation of CXCL12, and downregulation of dendritic cells among others and result in impaired immune mechanisms and protection of tumor cells [23]. BTK affects cell growth, survival, and migration of tumor cells via activation of CXCR4 [39]. Myeloid-derived suppressor cells (MDSCs) are a unique population of myeloid cells that expand during cancer and suppress T-cell responses thus allowing tumors to thrive and evade the immune system [40]. Because of its important role within tumor microenvironments, MDSCs are heavily sought out as the target of oncologic therapies. BTK is used by MDSC for TLRs, and as such is one of the reasons why BTK inhibitors are successful against hematologic malignancies [5].

## 6. BTK and Microglia

Microglia are the most common immune cells which comprise 10–20% of all glial cells in the CNS and are phenotypically and functionally related to cells of the monocyte/macrophage lineage. These cells are considered resident macrophages of CNS, much like the Kupfer cells in the liver or dendritic cells in the skin. They are involved in pathogen clearance and inflammatory and immune reaction to injury. It appears that microglia contribute to both inflammatory and reparative functions in MS, however, the timings of these different roles is somewhat unclear [41]. Microglia function is heterogenous with variable phenotype, depending on the CNS microenvironment, microglia can trigger neurotoxic pathways leading to neurodegeneration, or promote neuroprotection, downregulate inflammation, and stimulate repair [10,42,43]. In MS and related experimental models, the role of glial cells including microglia and astrocytes has been recognized recently. Activated microglia, particularly the M1 phenotype, are pro-inflammatory and contribute to demyelination and axonal injury. They function as antigen presenting cells (APCs) and secrete pro-inflammatory cytokines. The M2 phenotype, on the other hand, are neuroprotective, regulate immune function and promote remyelination and repair through the expression of anti-inflammatory molecules, phagocytosis of myelin debris, and production of growth factors important for oligodendrocyte proliferation [10,43,44,45]. M2 microglia are arginine-rich, and unlike the M1 phenotype that metabolize glucose anaerobically, the M2 microglia engage the mitochondria and generate more energy since they utilize the citric acid cycle for better utilization of glucose and fatty acids.

Chronic proinflammatory microglial activation in the CNS is believed to be one of the pathogenic mechanisms contributing to disability progression in patients with progressive disease [43,46]. Microglial activation and interaction with astrocytes, macrophages, and T cells in the CNS induce chronic inflammation behind a closed blood–brain resulting in release of proinflammatory cytokines, reactive oxygen species, nitric oxide causing damage to oligodendrocyte and myelin [47,48,49]. Inhibition of microglial activation results in reduction in inflammatory mediators in the CNS and attenuates disease in animal models of MS [50]. Within the CNS, BTK is mainly expressed in microglia, and to a lesser extent in astrocytes; inhibition of BTK may have potential impact on the function of microglia [8,9]. In animal models, the level of BTK expression has been shown to increase after demyelination, with inhibition of BTK favoring remyelination. [8] Neuropathological studies have shown higher expression of BTK in microglia derived from lesions of patients with progressive MS and in demyelinating mouse models [8,51]. BTKi have the potential to affect disease progression and disability in MS by targeting microglia, although it remains unclear whether the effect is mediated through suppression of inflammation and/or promotion of neuroprotection, and whether the effect is mediated by microglia, astrocytes, or both [8,10]. Nevertheless, the mechanism is believed to be complementary to the well-known effect of BTK inhibition on peripheral B-cell modulation and blockage of pro-inflammatory macrophages differentiation [8]. Targeting B cells via anti-CD20 antibodies in MS is a well-established treatment strategy, however, with limitations of low blood–brain barrier (BBB) penetration, modest effect on disability progression, and long-term safety concerns due to non-selective B-cell depletion resulting in immune suppression or reduced immune surveillance [10]. Therefore, the recent research has focused on BTK inhibition as an alternative approach to suppress B-cell activation. BTK inhibition using a potent, highly selective, rapidly reversible, BBB-penetrant small molecule BTKi could potentially reduce disease activity and slow disease progression by suppressing neuroinflammation mediated by activated microglia in the CNS, complementing its peripheral anti-inflammatory effect through B-cell suppression/modulation [9].

## 7. Development and Current Use of BTK Inhibitors as Cancer Therapy

Kinase pathways are involved in tumor cell survival and aberrant activation of protein kinases, which result in alterations in cellular proliferation, survival, motility and metabolism, as well as angiogenesis and evasion of the anti-tumor immune response [52]. Given the critical role of BCR in the proliferation and survival of cancer cells in a variety of B-cell malignancies, disrupting BCR signaling pathways involved in tumor cell survival through targeting its components such as BTK seems to be a good therapeutic approach. Additionally, BTK is present in other cells that contribute to regulation of tumor microenvironment including dendritic cells, macrophages, myeloid derived suppressor cells and endothelial cells. BTK inhibitors target cell homing, signaling, maturation, and survival in tumor microenvironments, resulting in apoptosis of cells in B-cell malignancies.

The majority of BTKi target ATP binding sites and can be divided into irreversible or reversible types, depending on their binding site and mode [53,54]. Small molecule covalent irreversible BTK inhibitors cause enzyme inactivation and BCR signal transduction blockage by binding with nucleophilic cysteine 481 residue (Cys-481) thiol group within the ATP-binding site of the kinase domain of BTK [55]. The reversible BTKi (e.g., fenebrutinib) do not bind the C481 site and block B-cell signaling by binding the BTK protein with a reversible non-covalent bond, thereby providing an effective alternative in patients with resistance following prior therapy with covalent BTKi [56]. Iibrutinib is a potent, small molecule irreversible BTKi with wide-ranging anti-tumor effects including interference with B-cell activation, reversal of CD4+ Th2 cells on Th1 cells shift, and inhibition of MDSCs expressing BTK (resulting in reduction in immune escape cytokines production) and MDSC migration and proliferation [23,57]. It was the first of its class of oral covalent BTKi to demonstrate clinical efficacy in patients with chemo-refractory B-cell malignancies and was approved in 2013 for treatment of mantle cell lymphoma (MCL) [58,59,60,61,62]. Currently, ibrutinib is approved for a variety of B-cell malignancies including MCL, chronic lymphocytic leukemia, lymphoplasmacytic lymphoma, Waldenström macroglobulinemia, small lymphocytic lymphoma, and marginal zone lymphoma [58]. However, it is a non-selective inhibitor and its off-target activities on other kinases including epidermal growth factor receptor (EGFR), SRC, and other Tec family kinases can result in side effects such as atrial fibrillation and bleeding [58,60,63,64]. Acquired resistance due to mutation of Cys-481 residue of BTK resulting in disruption of its covalent interaction has been reported [65].

The side effects related to off-target activity and acquired resistance to conventional irreversible BTKi (e.g., ibrutinib) prompted development of a novel second-generation BTKi with different organic structural orientation and binding profile, and better selectivity. Acalabrutinib was the first one of the second-generation highly selective irreversible BTKi to be approved for treatment of MCL in 2017 based on the results of a phase II clinical trial which showed good efficacy in patients with relapsed or refractory MCL [58,66,67]. It has greater selectivity with specific inhibition of certain kinases and cell signaling pathways compared to ibrutinib resulting in better oral bioavailability and less off-target side effects [66,67]. Tirabrutinib and zanubrutinib are highly selective BTKi that covalently bind BTK allowing high-sustained occupancy of the target and promising safety profile; both have shown good efficacy and lower incidence of serious side effects in early phase studies in patients with relapsed/refractory CLL, WM, and MCL [58,68,69,70]. Zanubrutinib received accelerated approval by the FDA in 2019 for treatment of refractory MCL [71]. Currently, there are several studies underway to study the safety profile of second generation BTKi including head-to-head trials comparing different agents [58].

Given the success of BTKi in various B-cell malignancies, these agents are being explored as treatment options for solid tumors such as primary prostate, ovarian, colorectal, and brain cancers. The effect of BTKi on tumor microenvironment and kinase inhibition result in cancer cell movement into peripheral circulation, where they may be more susceptible to traditional chemotherapeutic drugs. This has prompted exploration of combination of more potent and highly selective second-generation BTKi with proteasome inhibitors, anti-CD-20 antibodies, and other targeted agents in B-cell malignancies [57,72].

## 8. Role in Autoimmune Disorders

There has been recent interest in exploring BTKi for treatment of immune-mediated disorders. B cells have diverse roles in autoimmunity including cytokine and autoantibody production and interaction with T cells. Autoimmune diseases such as SLE and Sjogren’s syndrome (SS) involve B-cell dysfunction characterized by production of autoreactive antibodies and pro-inflammatory cytokines and the presentation of autoantigens to autoreactive T cells. BTK is involved in several signaling pathways in B cells and other immune cells including monocytes and macrophages downstream of BCR, Fc receptor, and TLR, thereby playing a central role in B-cell function, intrinsic tolerance, and immune complex activation of myeloid cell derivatives [5,6,31,35,73]. There is strong evidence for BTK being an important driver of B-cell–T-cell interaction, promoting autoimmunity. Increased BTK protein expression in patients with systemic autoimmune disease appears to correlate with autoantibody production [74]. By virtue of their ability to modulate aberrant BCR signal responsiveness and controlling immune tolerance, BTKi seem to be a promising targeted therapy option as they do not cause significant immunodeficiency, an issue with traditional B-cell depleting therapies. As BTK is expressed in multiple immune cell types, it may provide additional therapeutic benefit beyond B-cell depletion or blockage provided by traditional B-cell-based therapies by affecting the innate component mediating the disease pathogenesis.

The initial evidence of the protective effect of BTK inhibition against autoimmunity comes from animal studies with observations that BTK inhibition reduced disease symptoms whereas transgenic BTK overexpression induced systemic autoimmunity in mice [74,75]. BTK inhibition in experimental autoimmune diseases in preclinical models results in reduced disease activity, notably in mouse models of collagen-induced arthritis and anti-glomerular basement membrane glomerulonephritis, while increased BTK expression and phosphorylation in patients with systemic autoimmune disease correlated with autoantibody production [74,76]. BTK regulation in mouse models has been shown to be important for B-cell tolerance; BTKi have been shown to be effective in several systemic autoimmune mouse models like RA and SLE [7,77,78,79]. Although BTKi have shown excellent BTK suppression in animal models, the limited selectivity of highly potent BTKi (e.g., ibrutinib and acalabrutinib) can result in off-target activity and serious side effects [80]. This may be acceptable for cancer treatment given the disease severity and limited time window of treatment, however, limits the use of BTKi for chronic treatment of autoimmune diseases. Given the strong evidence of a pathogenic role of BTK in autoimmunity, several small-molecule BTKi with higher selectivity and low potential for off-target related adverse effects are being explored as a therapeutic option and are in the early phase of development for treatment of autoimmune diseases including MS, RA, pemphigus, and SLE [68,70,81].

## 9. Systemic Lupus Erythematosus (SLE)

SLE is a multisystem autoimmune disease with a pathogenesis involving autoreactive B cells and macrophages with deposition of immune complexes, complement activation, infiltration of immune cells, and increased cytokine production due to loss of immune tolerance [82]. Given the important role of BTK in BCR and Fc γ receptor signaling and activation of monocytes and macrophages, it presents a more selective target for treatment of SLE, better than systemic immunosuppression which is associated with serious off-target effects. The role of BTK and efficacy of BTKi have been studied in murine lupus models. In an experimental FcR-dependent, Ab-mediated model of glomerulonephritis, administration of PF-06250112 (an oral BTK inhibitor) at the time of induction reduced proteinuria, supporting the role of inhibition of the BTK signaling pathway in limiting the disease development [83]. In another study in transgenic mice, overexpression of BTK in B cells caused increased autoantibody production and lupus-like disease [79]. In an animal model of SLE, a novel, highly selective oral BTKi evobrutinib demonstrated effect on disease severity by inhibiting B-cell activation and autoantibody production and normalization of T- and B-cell subsets [7]. A preclinical study of ibrutinib in mouse model of lupus nephritis showed reduction in anti-nucleosome antibodies and anti-histone antibodies, but not anti-dsDNA antibodies, and improvement in some clinical manifestations including renal disease [84]. Fenebrutinib is a small molecule, highly selective, non-covalent, reversible BTKi which was well tolerated with no serious adverse effects or dose-limiting side effects in human volunteers [85]. In a study of SLE patients without renal involvement, treatment with fenebrutinib resulted in dose-dependent reduction in CD19 + B-cell counts and double-stranded DNA autoantibodies levels without significant clinical response [86]. The strong preclinical evidence of efficacy of BTKi in SLE is encouraging, however, it remains to be seen whether that will translate into clinical success.

## 10. Rheumatoid Arthritis (RA)

RA is another B-cell-mediated autoimmune disease where inhibiting BTK in macrophages and B cells may be a promising therapeutic target for RA. In the RA mouse model, efficacy of BTKi evobrutinib was demonstrated by action on B cells and innate immune cells although antibody reduction was not seen [7]. Spebrutinib is an orally administered, small-molecule BTK inhibitor recently explored for treatment of RA. Clinical pharmacodynamics were studied in 47 patients with active RA on background methotrexate therapy randomized to oral spebrutinib 375 mg/day or placebo. The drug was well tolerated and showed a trend for clinical improvement associated with lower increases in CD19^+^ B cells and greater decreases in CXCL13 and MIP-1β from baseline to week four [87]. When primary human B cells, T cells, natural killer cells, macrophages, dendritic cells, basophils, and osteoclasts were treated with spebrutinib in vitro, spebrutinib inhibited B-cell proliferation, and reduced lymphoid and myeloid cytokine production and osteoclast activity [87]. In a small in vitro study of patients with RA and SS, increased BTK activity was detected in memory and naïve B cells in anti-citrullinated protein antibody positive patients while BTK protein was increased in B cells in the majority of patients with SS with the levels correlating with serum rheumatoid factor levels and parotid gland T-cell infiltration [74]. A phase II open label study of selective, reversible, and non-covalent BTKi fenebrutinib demonstrated dose-dependent reductions in the rheumatoid factor with comparable efficacy to adalimumab in RA patients with inadequate response to methotrexate [88].

## 11. Multiple Sclerosis (MS)

B-cell depleting therapies have proven to be a highly effective treatment strategy in MS. B cells act as APCs and interact with T cells via release of cytokines and/or expression of costimulatory molecules and are considered key players in the pathogenesis of MS. Following favorable results of phase II study of rituximab and phase III trials of ocrelizumab and ofatumumab, several B-cell depleting therapies have been approved by the FDA and are now widely used for treatment of MS [11,12,13]. Great therapeutic success of these agents has sparked considerable interest in other novel therapies with different mechanism of targeting B cells due to safety concerns associated with long-term non-selective B-cell depletion (resulting in humoral deficiency), limited BBB penetration, and modest effect on disease progression. Among these are B-cell targeted therapies that do not deplete B cells but alter their ability to develop into the more mature forms capable of antibody production, cytokine secretion and antigen presentation. The interest in BTKi as a rational, non-depleting B-cell targeted therapeutic option for treatment of MS stems from BTK’s role in BCR and other innate immune cell intracellular signaling pathways, expression in various cells of hematopoietic origin and in CNS-resident innate microglia and astrocytes. BTKi’s anti-inflammatory effects through selective modulation of peripheral autoreactive B cells (sparing the normal B cells), attenuation of B-T-cell interaction and T-cell pro-inflammatory responses, and suppression of pro-inflammatory microglia, make them a better therapeutic option compared to traditional anti-CD20 therapies [89].

## 12. Presumed Mechanism of Action of BTKi in MS

Since BTK is expressed in myeloid cells that span innate and adaptive immunity, inhibitors of this enzyme can have a multipronged effect on immune modulation. The following sections describe the potential effects of BTKi on different steps of the immunopathogenic cascade of MS; the specific checkpoints where these therapies may alter the immune response are schematically represented in Figure 3.

### 12.1. Checkpoint 1. Antigen Presentation and Breakdown of Immune Tolerance

Presumably, this is the first event that leads to autoimmunity and these events occur in the systemic compartment outside the CNS. Programmed autoreactive B cells function as APCs to T cells, and it is speculated that this B cell–T cell interaction is disturbed by the B-cell depleting therapies leading to the beneficial effects observed by these agents in a number of autoimmune disorders including MS. While BTKi may inhibit B-cell proliferation mediated through antigen engagement of the BCR, it is not known if this decreased proliferation will lead to decrease in antigen presentation by B cells to T cells. However, using the BTKi evobrutinib in an animal model of experimental immune-mediated demyelination, it was evident that antigen-mediated B-cell signaling was impaired and the B cells were less efficient in programming T cells to become encephalogenetic [90]. This same study also identified decreased antigen presentation by the reduction in class 2 molecules on antigen-presenting B cells. In addition to impaired antigen presentation, the effects on monocytes and macrophages can result in downregulation of Fc gamma R III function and reduction in the secretion of pro-inflammatory lymphokines, TNF alpha, IL-1β and IL-6 as shown in a study of BTKi in collagen-induced arthritis [91].

### 12.2. Checkpoint 2. CNS Migration of Autoreactive Cells

Autoreactive cells, particularly pathogenic B cells, have unrestricted access to the CNS. They enter the CNS, undergo reactivation and proliferation to become resident CNS cells in the meningeal lymphoid follicles. In vitro studies have identified that human endothelial cells treated with BTKi resist transport of BTK-bearing cells across the endothelial barrier. In a model of prostate cancer, Ibrutinib was shown to be able to block production of MMP2 and MMP9, both enzymes important in the dissolution of the basement membrane, an important step in the movement of lymphocytes across the BBB [92]. B-cell interaction with stromal cells can be disrupted by BTK inhibition, another mechanism by which migration of B cells across endothelial surfaces can be reduced [93].

### 12.3. Checkpoint 3. Secondary Activation of the Autoreactive Cells within the CNS

Following migration of autoreactive lymphocytes across the BBB to the perivenular spaces, these cells undergo secondary amplification by microglia that present the various CNS autoantigens to T and B cells to further amplify the immune response. These microglia, like their monocyte counterparts in the periphery, are BTK-bearing cells of the myeloid lineage. The pro-inflammatory M1 phenotype, constitutively express BTK and are amenable to polarization to the M2 phenotype following BTK inhibition. During treatment with BTKi, the M1 phenotype monocytes undergo apoptosis while the M2 phenotype enhance removal of apoptotic cells [94]. The arginine-rich M2 microglia are neuroprotective and do not participate in the secondary amplification needed for these cells to become autoreactive, thereby aborting any potential CNS injury.

### 12.4. Checkpoint 4. Astrocyte Modulation

It is well established that astrocytes regulate CNS inflammation by multiple mechanisms, and it is increasingly becoming recognized that astrocytes are pivotal cells that participate in CNS inflammation. Only 20% of all astrocytes express BTK, and in human gliomas the BTK expression appears to be restricted to the gamistocytic, or reactive astrocytes [95]. Inhibition of BTK in these cells can lead to significant amelioration of inflammation. In a lipopolysaccharide-induced model of neuroinflammation, astrocytes expressed significantly less activation and secretion of COX-2 and IL-1β cytokines tending to downregulate inflammation following BTK inhibition [96].

### 12.5. Checkpoint 5. Demyelination and Remyelination

When BTK and activated BTK (phosphorylated form, pBRK) were examined by immunohistochemistry in a model of lysolecithin-induced demyelination of organotypic cerebellar slices cultures, a significant increase in BTK activity from resting state was identified after demyelination, almost 8-fold greater activity as compared to the resting state. About 75% of this activity was observed in the microglia and 25% in astrocytes. BTK inhibition resulted in accelerated remyelination and identified a potential role for this approach in improving return of myelin following demyelination [8].

### 12.6. Checkpoint 6. Axonal Injury

Axonal transection is an integral part of the pathology of early MS and up to 11,000 axons are estimated to be transected per mm [3] of demyelination [97]. Microglia has been implicated in the process of axonal transection, and early studies that identified such transection established a role for microglia. As indicated, pro-inflammatory microglia are involved in early CNS injury, and inhibitors of BTK would be expected to also have a favorable effect on also limiting injury to the axons. A recent study from this group has identified such an effect from a brain-penetrant BTKi in reducing CNS inflammation [98].

## 13. Pre-Clinical Studies of BTKi in MS

Most of the experience with BTKi in MS comes from pre-clinical studies which have shown that targeting BTK can counter B-cell pro-inflammatory responses. Small molecule tyrosine kinase inhibitors (imatinib, sorafenib, and GW2580) were shown to prevent the development of disease and treat the established disease in a mouse model of MS [99]. In another study, a selective BTKi evobrutinib inhibited antigen-triggered activation and maturation of B cells, preventing development of encephalitogenic T cells and reducing disease severity in the mouse model of experimental allergic encephalomyelitis [21]. Additionally, BTK inhibition in animal models of demyelination and remyelination resulted in improved remyelination, suggesting BTKi as a potential strategy for promising myelin repair [8].

## 14. Clinical Studies of BTKi in MS

To date, two phase II trials of BTKi in patients with RMS (RMS) have been conducted, the first study examining evobrutinib and the second, more recent study involving tolebrutinib [100,101]. Evobrutinib was evaluated in a phase II clinical trial in patients with RMS and active secondary progressive MS (SPMS), a total of 267 patients were randomly assigned to receive placebo, evobrutinib (25 mg once daily, 75 mg once daily or 75 mg twice daily), or open-label dimethyl fumarate as a reference [101]. At 24 weeks, the placebo group was switched to evobrutinib (25 mg daily) while the rest of the groups were continued on the same medication for an additional 24 weeks. The primary endpoint was total number of T1 gadolinium enhancing lesion-count at weeks 12, 16, 20 and 24, the secondary endpoints included annualized relapse rate, disability scores from baseline, and safety. Treatment with evobrutinib 75 mg once daily resulted in dose-dependent reduction in MRI lesions from weeks 12 through to 24 compared to placebo while other doses did not show any efficacy on the primary endpoint. There was no effect on relapse rate or disability progression at any dose; the high dose evoburtinib group (75 mg once or twice daily) had higher frequency of elevated transaminase and lipase levels. When a post hoc analysis was done and serum neurofilament light chain levels were examined, a significant reduction in levels at weeks 12 through 24 as compared to placebo was identified in the 75 mg twice daily dose, further supporting a significant anti-inflammatory and protective role of BTKi [7,101]. Currently, two ongoing phase III trials (Evolution 1 and 2) of evobrutinib are enrolling patients with relapsing-remitting MS (RRMS) or active SPMS. The study goal is to compare the effect of evobrutinib with interferon β-1a on disease activity and disability.

Tolebrutinib is a highly selective, irreversible, covalent BTKi which crosses the BBB and has the potential to target inflammation in the periphery and CNS through modulation of microglia and brain-resident B cells, an effect that could be beneficial for patients with progressive forms of MS [41,102]. Additional therapeutic advantages of tolebrutinib include fast and reversible B-cell modulation (5–7 days until recovery) and once daily oral dose. Tolebrutinib was studied in a recent 16-week, phase 2b, randomized, double-blind, placebo-controlled, crossover, dose-finding study in 130 patients with RRMS and relapsing SPMS [100]. The treatment period was limited to 12 weeks to minimize exposure to placebo. Four doses of tolebrutinib were tested, 5 mg, 15 mg, 30 mg, and 60 mg administered once daily. The results demonstrated that treatment with tolebrutinib for 12 weeks led to a dose-dependent reduction in new gadolinium-enhancing lesions and new or enlarging T2 lesions with the 60 mg dose being the most efficacious. All doses were well tolerated; headache was the most common side effect. Currently, phase III clinical trials of tolebrutinib in patients with relapsing and progressive forms of MS (NCT04411641, NCT04458051, NCT04410991, and NCT04410978) are underway. Due to a few cases of drug-induced liver injury with tolebrutinib, phase III studies of tolebrutinib in the United States are currently on partial hold.

Another selective, non-covalent BTKi fenebrutinib that has been tested in other autoimmune disorders, is currently in phase III trials in Relapsing Remitting MS and primary progressive MS (PPMS) (NCT04586023, NCT04586010, NCT04544449) with first results expected in 2024 [103]. Fenebrutinib is a reversible inhibitor and probably needs relatively high CNS exposure to maintain therapeutic efficiency. Additionally, phase II randomized double-blind clinical trial of another second generation irreversible BTKi orelaburtinib in patients with RMS is ongoing (NCT047111148).

Other tyrosine kinase (TK) inhibitors are also being studied for treatment of MS. Masitinib is highly selective TK inhibitor with effects on KIT receptor which has a role in promoting cell growth, differentiation, and migration while not inhibiting other kinases associated with adverse effects, such as cardiotoxicity [104]. It is a novel oral therapy and represents a first-in-class TK inhibitor being studied as a treatment option for patients with PPMS and SPMS without relapses. Data from a small phase II pilot study showed a benefit with masitinib when compared to placebo, as demonstrated by an increase in patients’ MS functional composite (MSFC) score and stability of EDSS scores. The side effects were mild to moderate and included asthenia, leucopenia/lymphopenia, rash, and C+GI side effects [105]. A recent phase III, randomized, double-blind, 2 parallel group, placebo-controlled trial demonstrated a positive effect of masitinib 4.5 mg/kg/d on progression of disability, as measured by the EDSS, in patients with PPMS and relapse-free SPMS with no exacerbations in the last 2 years over a 96-week treatment period [106]. The side effects included diarrhea, nausea, rash, peripheral edema and neutropenia, consistent with masitinib’s known profile. 

Although there is good evidence for anti-inflammatory effects of BTKi mainly through indirect inhibition of pathogenic T-cell activation in animal models, it remains to be seen whether these will translate into clinical efficacy. In the absence of long-term clinical efficacy data on BTKi and head-to-head comparison with anti-CD-20 monoclonal antibodies, it is difficult to comment on the relative efficacy of these therapies in MS. There is a possibility that BTKi may not be as effective as classic anti-B-cell therapies based on indirect comparison of phase II data (12-week rituximab showing early and robust effect on MRI activity and relapse rate in contrast to lack of clinical efficacy in similar evobrutinib study). However, the results of phase III clinical trials are needed to draw any real conclusions about their long-term efficacy. The ease of administration and CNS penetrability of BTKi are attractive qualities compared to anti-B-cell therapies and their role in treatment of MS in the future may include using them as maintenance therapy following induction with anti-B-cell therapies to minimize effects of prolonged B-cell depletion, most notably, hypogammaglobulinemia.

Conclusion (Table 1): Bruton’s tyrosine kinases play a key role in B-cell and other myeloid cell development, proliferation, and function. BTK inhibition has been established as an effective strategy for the treatment of B-cell leukemia and other hematological malignancies. Small molecule BTKi are now being studied as treatment options for a variety of autoimmune diseases with aberrant B-cell function including lupus, rheumatoid arthritis, and MS. Advantages of BTKi as a treatment option for MS over traditional B-cell depleting monoclonal antibodies include less potential risk of chronic immunosuppression and dual effects via peripheral modulation of B- and other innate immune cell signaling pathways and central inhibition of CNS-resident, innate microglia and astrocytes. Whether the BTKI will be more efficacious than anti-B-cell monoclonal antibodies remains to be established. Furthermore, there are potential safety concerns related to non-selective inhibition of tyrosine kinase resulting in off-target effects. Additionally, the long-term immune effects of BTKi remain unknown especially the risk of opportunistic infections and response to vaccines. Several phase III studies ongoing to evaluate the safety and efficacy of BTKi for treatment of MS will help address some of these questions and concerns.

After initial antigen-binding by surface immunoglobulin, BCR is activated along with autophosphorylation of its signal transduction molecules CD79A/CD79B, followed by recruitment of LYN (Lck/Yes novel kinase) and SYK (spleen tyrosine kinase). BTK is then activated by SYK-medicated phosphorylation and autophosphorylation along with its partner scaffold protein BLNK, both in turn mediate activation of PLC-γ2. Activation of PLC-γ2 results in downstream generation of IP3 and DAG leading to increased intracellular Ca^2+^ (iCa^2+^) and activation of PKC enzymes. The ultimate downstream effects include activation of NFAT and NF-kB factors and BTK, AKT, and mTOR pathways, resulting in increased transcription of genes involved in control of survival, migration, and proliferation of B cells.

Figure 3 Potential checkpoints where cells expressing BTK are affected by BTK inhibitors to potentially influence lesion formation in Multiple Sclerosis.

The systemic compartment (venule) is separated from the CNS by the blood–brain barrier. Sequentially the checkpoints where BTK inhibitors could potentially suppress the lesion formation are:

Checkpoint 1. Breakdown of immune tolerance. The dendritic cells process the environmental trigger, potentially a virus or other pathogen, and sensitize T cells to become autoreactive by molecular mimicry. Autoreactive CD4, CD8 and B cells are generated. BTKi can reduce limit activation of dendritic cell or B cells, both of which express BTK.

Checkpoint 2. Migration of autoreactive cells from the systemic to the CNS compartment across the BBB. Endothelial cells express BTK and can be potentially modulated to reduce such transit.

Checkpoint 3. Secondary expansion of autoreactive cells within the CNS compartment. Antigen presenting microglia express BTK, and inhibitors of BTK can abrogate this necessary step for demyelination.

Checkpoint 4. Reactive astrocytes (gemistocytes) are potent mediators of CNS inflammation and express BTK and may be amenable to modulation by BTKi.

Checkpoint 5. Demyelination. Pro-inflammatory M1 microglia may mediate demyelination by secretion of pro-inflammatory cytokines, which are known to mediate myelin injury.

Checkpoint 6. Axonal transection. M1 microglia are also known to cause axonal injury with retrograde Wallerian degeneration and death of the neuron (brown-shaded soma).

## Figures and Tables

**Figure 3 jcm-11-06139-f003:**
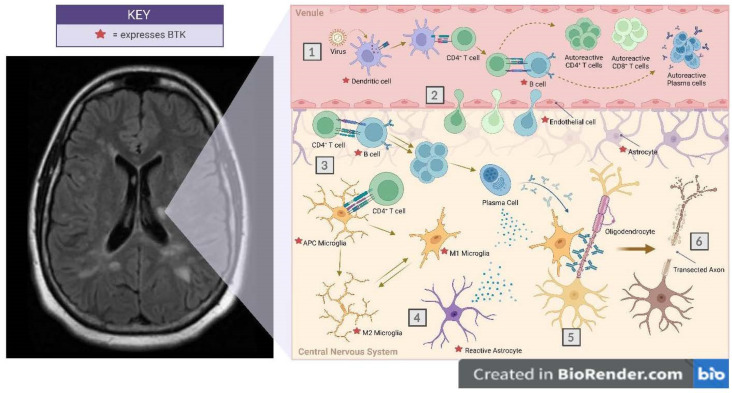
Systemic and CNS events leading to the formation of the MS plaque are schematically demonstrated, with the various cells involved in the occurrence of the inflammatory demyelinating lesion. Cells expressing BTK are tagged with the red star.

**Table 1 jcm-11-06139-t001:** Phase 2 and 3 Clinical Trials of BTK and other TK inhibitors in MS [100,101,106].

Drug	BTK Binding	Phase II Studies	Phase III Studies
Evobrutinib	Covalent, irreversible	RMS and active SPMS, reduced enhancing lesions compared to placebo	RMS (evolutionRMS1 and RMS2), evobrutinib vs. teriflunomide (NCT04338022, NCT04338061)
Tolebrutinib	Covalent, irreversible	RRMS and relapsing SPMS, reduced new active brain MRI lesions	RMS, tolebrutinib vs. teriflunomide PPMS and non-relapsing SPMS, compared to placebo (NCT04411641, NCT04458051, NCT04410991, and NCT04410978)
Fenebrutinib	Non-covalent, highly selective, reversible		Relapsing MS and active SPMS, fenebrutinib vs. teriflunomide Primary progressive MS, fenebrutinib vs. ocrelizumab (NCT04586023, NCT04586010, NCT04544449)
Remibrutinib	Covalent irreversible		RMS, remibrutinib vs. teriflunomide (NCT05147220, NCT05156281)
Orelabrutinib	Covalent, irreversible	Randomized, double-blind, placebo-controlled study ongoing (NCT04711148)	
Masitinib	Selective tyrosine kinase inhibitor		Randomized, double-blind, 2 parallel-group, placebo-controlled trial in PPMS and non-relapsing SPMS showed positive effect on disability progression A confirmatory phase 3 study ongoing, PPMS or non-relapsing SPMS, compared to placebo (NCT05441488)

## Data Availability

Not applicable.

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
