# Peer review of "Bruton’s Tyrosine Kinase Inhibitors: The Next Frontier of B-Cell-Targeted Therapies for Cancer, Autoimmune Disorders, and Multiple Sclerosis"

_jcm, 2022, doi:10.3390/jcm11206139_

Round 1

Reviewer 1 Report

A review of the manuscript titled „Bruton’s Tyrosine Kinase Inhibitors: The next frontier of B cell 1 Targeted therapies for cancer, autoimmune disorders, and MS” by Neeta Garg, Elizabeth Jordan Padron, Kottil Rammohan, Courtney Frances Goodman to Journal of Clinical Medicine.

The authors outline the role of Bruton's tyrosine kinase (BTK) in the body, as well as the potential medical application of inhibitors of this enzyme. The work is well thought out, each study is well reasoned. However, there were some minor understatements in it. Please reconsider the following:

1.       What does mean „aka”  in line 88?

2.       Including clinical trials involving BTKI in the Table would be beneficialble. Admittedly, the authors have included some information on this subject in the text, but it would be more readable to include this information in the form of a Table.

Author Response

  1. What does mean „aka”  in line 88?

The abbreviation “aka” in line 78 has been defined (also known as)

  1. Including clinical trials involving BTKI in the Table would be beneficialalbe. Admittedly, the authors have included some information on this subject in the text, but it would be more readable to include this information in the form of a Table.

Thank you for this valuable suggestion, we have modified table 1 to include MS phase 2 and 3 study details

Reviewer 2 Report

In this review, authors summarized emerging literature on the role of Bruton’s tyrosine kinase (BTK) in B-cell receptor signaling and the development of BTK inhibitors for the treatment of B-cell malignancies, multiple sclerosis (MS) and other autoimmune disorders. Since BTK played significant role in the many signaling process including cell survival, proliferation, cell migration, and therapy resistance that drive B-cell malignancies and other autoimmune disorders, the proposed review is timely and clinically significant. Addressing the following suggestions will improve the manuscript.

1)     Even though, MS is a well-known term in this field, it is ideal to mention multiple sclerosis in the review title to avoid further confusion.  

2)     BTK inhibitors should be mentioned as BTKi instead of BTKI throughout the manuscript.

3)     It is an elegantly written review. Mentioning some more pre-clinical studies will strengthen the role of BTK inhibition in cancer therapy.

4)     In table 1, the first column is empty. It can be removed if not required or mention NCT number for the respective clinical trials.

5)     Figure 3 is pixelated. Improve the quality of the image.

Author Response

Reviewer #2:

In this review, authors summarized emerging literature on the role of Bruton’s tyrosine kinase (BTK) in B-cell receptor signaling and the development of BTK inhibitors for the treatment of B-cell malignancies, multiple sclerosis (MS) and other autoimmune disorders. Since BTK played significant role in the many signaling process including cell survival, proliferation, cell migration, and therapy resistance that drive B-cell malignancies and other autoimmune disorders, the proposed review is timely and clinically significant. Addressing the following suggestions will improve the manuscript.

  • Even though, MS is a well-known term in this field, it is ideal to mention multiple sclerosis in the review title to avoid further confusion.  

The article title has been modified to include multiple sclerosis

  • BTK inhibitors should be mentioned as BTKi instead of BTKI throughout the manuscript.

BTKI has been replaced with BTKi throughout the manuscript

  • It is an elegantly written review. Mentioning some more pre-clinical studies will strengthen the role of BTK inhibition in cancer therapy.

We really appreciate the comment, however, feel that the information presented in the previous sections on role of BTK in B cell development and signaling pathways and its effect on other cells provide sufficient background on evolution of BTKi to the current established role in cancer therapy.

In table 1, the first column is empty. It can be removed if not required or mention NCT number for the respective clinical trials.

We have modified the table to remove first empty column, NCT trial information has been added to the last column

5)     Figure 3 is pixelated. Improve the quality of the image.

 The figure has been replaced with a higher resolution better quality image